# Nutritional assessment of adolescents: A cross-sectional study from public schools of North India

**Sandeep Kaur**[1,2,3]*, **Rajesh Kumar**[1], **Manmeet Kaur**[1]

**1** Department of Community Medicine and School of Public Health, Post-Graduate Institute of Medical Education and Research, Chandigarh, India, **2** Department of Surgery and Cancer, Faculty of Medicine, Imperial College London, London, United Kingdom, **3** Indian Institute of Public Health-Delhi, Public Health Foundation of India, New Delhi, India

* sandeep.kaur@imperial.ac.uk

## Abstract

### Background

Technological advancements and globalization have shifted dietary behaviours, contributing to increased chronic disease prevalence in Low- and Middle-Income Countries (LMICs) like India. Adolescents are particularly vulnerable due to these changes, which can impact their lifelong health. This study aimed to assess the nutritional status of adolescents in public schools in Chandigarh, India.

### Methodology

Conducted as part of a cluster randomized control trial, the study used two-stage random sampling to select 12 schools and eighth-grade classes, recruiting 453 adolescents aged 10–16 years. Nutritional status was evaluated through dietary behaviour assessments, anthropometric measurements, and 24-hour urinary salt-level analysis. Dietary patterns were recorded using two 24-hour recalls, and analyzed with PURE study software based on 2010 Indian dietary data from ICMR-NIN. Anthropometric measures followed standardized protocols, and salt levels were assessed in laboratories.

### Results

The mean age of the adolescents was 13.06 years, with 55% being boys. Among them, 32% had high salt intake, and 55% had high sugar intake. Additionally, 90% had low fruit intake, and 83% had low vegetable intake. The adolescents were deficient in several macro and micronutrients, including energy, fats, fibre, iron, zinc, iodine, riboflavin, and vitamins B-6 and B-12. A higher proportion of boys (10%) were classified as thin compared to girls (2%), while a greater proportion of girls (36%) had abdominal obesity. In contrast, a larger proportion of boys (23%) were severely acutely malnourished. Nearly all adolescents exhibited high urinary excretory salt levels.

**Data Availability Statement:** The entire dataset used for this manuscript's analysis can be accessed at the repository with doi: 10.6084/m9.figshare.24902559. This dataset consists of

'minimal data set' with related metadata to replicate the reported study findings in their entirety.

**Funding:** Partial funding was awarded to MK with project ID 5562 for undertaking laboratory tests of 24-hour urine samples from The George Institute for Global Health. The funders had no role in study design, data collection and analysis, decision to publish, or preparation of the manuscript.

**Competing interests:** The authors have declared that no competing interests exist.

## Conclusion

Most adolescents exhibited dietary risk factors, including high salt and sugar intake, along with low consumption of fruits and vegetables. Many were deficient in various macro and micronutrients, with the coexistence of both thinness and obesity. Regular nutritional assessments in schools are essential to address the dual burden of undernutrition and over-nutrition. Furthermore, health-promoting interventions should be developed within school settings to encourage healthy dietary practices.

## Introduction

Technological advancements, easy availability, and tempting advertisements of processed foods allure young people, especially adolescents, to consume energy-dense foods with low nutritional value, finally leading to a buildup of unhealthy dietary behaviours in adulthood [1].

Healthy dietary behaviours have been as low as 40% worldwide, with scenarios marginally better in high-income countries (HIC) compared to low-and-middle-income countries (LMICs) [2]. Among LMICs, India's situation is gruesome, with only 10.6% of Indians following healthy dietary behaviours [3]. India is in the midst of a global transition, wherein it faces the double burden of both under-nutrition and over-nutrition [4, 5]. A recent NFHS-5 survey (15–49 years) revealed that 23% of women and 20% of men were underweight [6]. In comparison, 21% of women and 19% of men were overweight or obese [6]. These findings signify the co-existence of underweight and overweight among the Indian population [6].

The rising prevalence of malnutrition has contributed to the epidemiological transition, marked by an increase in chronic diseases and associated health burdens [4, 7]. As, in the same NFHS-5 survey, 14% of women and 16% of men had high blood glucose levels and 21% of women and 24% of men had high blood pressure levels [6].

The behaviour of unhealthy dietary practices leading to malnourishment emerges during early adolescence [1]. When these behaviours are not addressed timely, the early year of malnourishment becomes one of the most significant risk factors in the occurrence of chronic diseases in later life [7, 8]. Adolescence is a transition phase of human life, both physically and socially [8]. At this age, the quality of nutrition is one of the important factors leading to the holistic growth of adolescents into healthy adults [9].

It is the need of the hour to regularly assess nutrition behaviour during early adolescence to address the issue of malnutrition and keep a check on one of the risk factors of chronic diseases [1, 7]. Owing to expensive, lifelong treatments and loss of productive life years due to the increasing premature deaths and disabilities related to various chronic diseases in India, it is recommended to focus more on promotion and prevention than curative measures [10]. Therefore, as a part of a larger comprehensive behaviour-change intervention, the present cross-sectional study aims to assess the nutrition status and anthropometric measures of early adolescents studying in public schools in a city in North India [11].

## Methods

### Study design

The present analysis is based on a cross-sectional survey carried out as part of a broader cluster randomized control trial to assess the impact of a school-based health promotion intervention package in reducing the behavioural risk factors associated with chronic diseases [12]. The

survey was conducted among adolescents studying in the public schools of the North Indian city of Chandigarh, India from May 2018 to September 2019.

## Study setting

The study was conducted in public schools as the Director of Public Instruction did not allow enrolment of private schools in the study. Data from the Department of School Education, Chandigarh, revealed that 106 public schools had eighth-grade classes, with an enrollment of 10,790 adolescents. Most schools had four to five sections in 8[th]-grade class. Each section of the class had about 30 adolescents.

## Participants

Based on the knowledge that behaviours leading to malnourishment emerge during early adolescence and analysis of the school curriculum, consenting adolescents between the ages of 10–16 from eighth grade were included in the present study [8]. An additional inclusion criterion was the adolescents not intending to leave the city for another year from the recruitment date.

## Sample size and sampling

Details on the sample size calculation are mentioned in the protocol paper of the broader CRCT study. Broadly, the required number of clusters for 80% power for each primary outcome was calculated separately using expected means or proportions for the control arm, intervention effects, and coefficient of variation values based on data for adolescents from recent studies [13–17]. Out of all risk factors, the maximum 'c' value (n = 12) was considered the sample size of the number of clusters for the study [12]. From the calculated sample of 12 clusters (schools), and cluster size of 30 adolescents, a final sample size of 360 adolescents was estimated.

## Outcome variables

Dietary intakes, anthropometric measurements and 24-hour excretory salt levels were the primary outcome variables. Nutritional status was assessed by estimating the dietary intake of salt, sugar, fruits and vegetables in g/d, along with macronutrients (energy, carbohydrates, proteins, fats and fibre), minerals (calcium, magnesium, iron, sodium, potassium, zinc and iodine) and vitamins (vitamin A, thiamine, riboflavin, niacin, vitamin- B6, folate, vitamin B-12 and vitamin C) intake. In addition to nutritional assessment, salt was also assessed through 24-hour urine samples in the form of excretory-urinary salt levels. The null hypothesis posited that all adolescents would have adequate macro- and micronutrient intake, along with normal body mass index (BMI), mid-upper arm circumference (MUAC), triceps skinfold thickness (TSFT), sub-scapular skinfold thickmness (SSFT), and urinary salt levels.Anthropometric measurements estimated the prevalence of severe thinness, thinness, overweight, obesity, abdominal obesity and severely acute malnourishment among the selected adolescents.

## Data collection tools

Nutritional status was measured through the administration of a 24-hour dietary recall questionnaire. This questionnaire was administered twice on non-consecutive days [18]. Data collected through the 24-hour dietary recall was analysed using the PURE study software. This software uses the National Institute of Nutrition (NIN) dietary guidelines for Indians to calculate the quantity of various macro and micronutrients from the quantity of various food items consumed by a person in a day [19, 20]. Socio-economic status (SES) was assessed using the

'Kuppuswamy 2018 scale' [21]. This scale uses information about the education and occupation of the head of the household and the household's total monthly income to score and rank participants into five categories of socio-economic status, i.e., lower, upper-lower, lower-middle, upper-middle, and upper [21].

Height was measured with UNICEF's standardised anthropometer to the nearest 0.1 cm. Weight was measured with a portable electronic weighing scale to the nearest 0.1 kg. Hip circumference (HC), waist circumference (WC), and MUAC were measured to the nearest 0.1 cm by using fibreglass measuring tape. TSFT and Sub Scapular SSFT were measured to the nearest 0.1 mm using Lange's skin-fold calliper.

For 24-hour urine samples, adolescents were advised to write the time of their voids and whether they had collected each void in the collection container or not for 24 hours using a structured proforma (S1 Appendix in S1 File).

## Data collection methods

The dietary and anthropometric data were recorded, and during the same period, from May 2018 to September 2019, a team of three members collected 24-hour urine samples.

Standard operating procedures were developed to maintain uniformity and standardization in sample collection. The self-administered 24-hour dietary recalls were recorded for all adolescents [18]. A team member was present to explain each question to adolescents before they recorded their responses. The first 24-hour dietary recall collection was during the first interaction with all the adolescents in the schools. The second 24-hour dietary recall collection was a week later. Anthropometric measures were recorded in a separate room after recording dietary data.

All anthropometric instruments were regularly calibrated as per standard requirements. The portable digital weighing scale was calibrated after every 20th reading. 'Lange's' skinfold calliper was checked for needle at 'zero' before every reading and was calibrated using a calibration block after every 20th reading. Fibreglass tape was used to measure hip, waist and mid-upper-arm-circumference as it is resilient to wear and tear and does not expand in warm weather conditions like an ordinary measuring tape [22]. Anthropometric measurements were recorded by a team member who had undergone training during the 2018 Comprehensive National Nutrition Survey (CNNS).The 24-hour urine samples were collected during house visits. During Saturdays' home visits, participants received instructions regarding the process of 24-hour urine collection on Sundays. Urine collection containers were provided to them. They were instructed to discard the first void of the morning and then start collecting the subsequent voids for the next 24 hours (S2 Appendix in S1 File). The urine samples were collected on Monday mornings and analysed in the Beckman Coulter AU 5800 analyzer (2020) using standardized laboratory methods to assess urinary salt levels on the same day (S2 Appendix in S1 File).

## Data management and analysis

The PURE study computer software was used to analyse the diet data and estimate the intake of macro and micro-nutrients (i.e., protein, carbohydrates, fat, vitamins, and minerals) [23]. From PURE software, 72 different macro and micro-nutrients can be estimated (S3 Appendix in S1 File). The widely used and accepted guidelines for Recommended Dietary Allowance (RDA) for the Indian populations from the Indian Council of Medical Research-National Institute of Nutrition (ICMR-NIN), Hyderabad (2020), were used to compare the dietary nutritional status of the adolescents estimated in the present study [24]. The twenty dietary nutrients considered essential for young adolescents' proper growth and health, as laid down

by ICMR-NIN, were used to assess the dietary nutrition for the present study [24]. The method of calculating salt, sugar, fruits and vegetables in g/d from 24-hour dietary recall is presented in the supplement (S4 Appendix in S1 File).

Adolescents' BMI was used to categorise adolescents into severe thin, thin, normal, over-weight and obese categories based on WHO growth reference percentiles that were age and sex-specific [25]. For waist-hip ratio (WHR), WHO cut-offs were used for all adolescents [26]. Age and sex-specific percentiles were used to categorise MUAC, TSFT and SSFT [27].

The complete quantitative data analysis was performed using the Statistical Package for Social Sciences (SPSS) version 21 and Microsoft Excel 2007 [28]. Descriptive statistical analysis includes the estimation of the sample mean (along with standard deviation) or proportions according to the variable types. As, the sample size was sufficient to test these hypotheses, unpaired-t or chi-square tests were used to explore potential differences between groups (e.g., age, gender and socio-economic status). ANOVA was used to test the statistical significance of continuous variables in bivariate analysis. P-value <0.05 was considered to be statistically significant.

The cut-offs to categorize all the adolescents into low, normal, high, very high and extremely high categories were used from literature to assess urinary salt excretion from 24-hour urine samples [29].

The authors have access to the participants' identifiable information, with access to this data limited to them. However, wholly de-identified data will be shared at an appropriate data archive for sharing purposes after three years of the end-line assessment data collection as this study is part of a larger cluster Randomized Controlled Trial.

## Ethical considerations

Ethical permission was obtained from the Institutional Ethics Committee of the Post-Graduate Institute of Medical Education and Research, Chandigarh (INT/IEC/2018/ 000082; Date. 22/ 01/2019). Permission was granted for the study in the schools by the Department of Education, Chandigarh, with letter no 1296-DSE-UT-S5-11(65)11-II. As the present study was part of a cluster randomised controlled trial, the study was registered in CTRI with no. CTRI/2019/09/ 021452.

Written assent of adolescents and written consent from their parents were obtained after informing them about the study purpose, data to be collected, estimated time for data collection, the confidentiality of the data, and risks involved before their enrolment in the study (S5 Appendix in S1 File).

## Results

Out of the 462 eligible adolescents from twelve randomly selected public schools, 453 consented to participate in this cross-sectional survey (S6 Appendix in S1 File). Out of selected adolescents, 55% were boys, 81% were 13–15 years old (81%), 87% belonged to the Hindu religion, 66% were from the general category, and 61% belonged to the upper-lower socio-economic status category (S7 Appendix: S1 Table in S1 File).

The consumption of salt, sugar, fruit and vegetables was also estimated separately (S8 Appendix: S2 Table in S1 File). The mean consumption of salt was 4.5 g/d which was within the RDA. The mean intake of sugar was 36 g/d. It was lower (34 g/d) among boys than girls (39 g/d).

The mean vegetable intake (193 g/d) was lower than the RDA of 300 g/d. The mean vegetable intake among boys was lower (169 g/d) than the girls (223 g/d). The average intake of fruits was 32.7 g/d, which is less than the RDA of 100 g/d (S8 Appendix: S2 Table in S1 File).

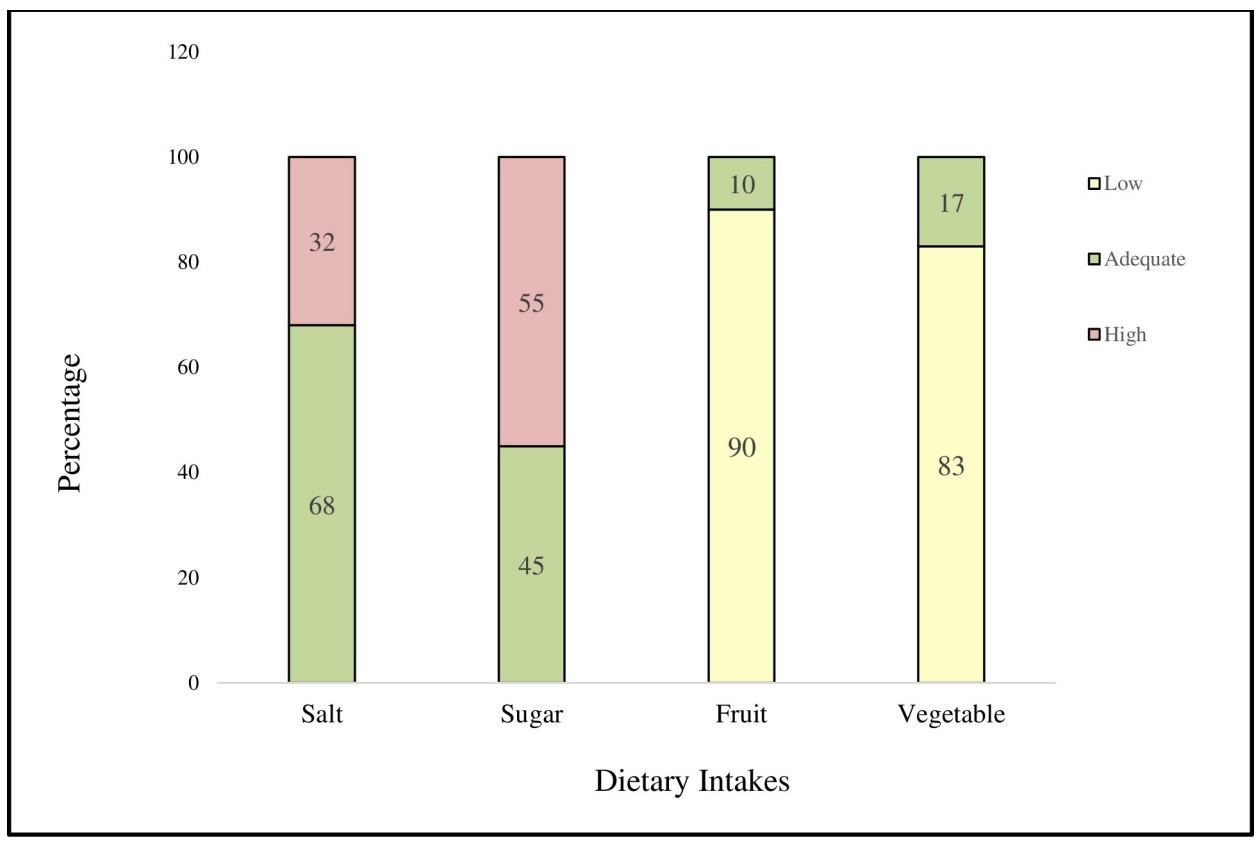

**Fig 1. Adequacy of dietary intakes of salt, sugar, fruits and vegetables among adolescents (n = 453). Cut-offs for dietary intakes|** adequate salt: ≤ 5 g/d, high salt: > 5 g/d, adequate sugar: ≤ 20 g/d for girls and 25 g/d for boys, high sugar: > 20 g/d for girls and 25 g/d for boys, low fruit < 100g/d, adequate fruit: ≥ 100g/d, low vegetable < 300g/d and adequate vegetable: ≥ 300g/d.

Thirty-two per cent of adolescents consumed more salt than the recommended dietary allowance (RDA), while 55% exceeded the RDA for sugar. At the same time, the consumption of fruits and vegetables was poor, with 90% and 83% of adolescents consuming low levels. Therefore, it was necessary to implement health promotion interventions to change these dietary behaviours (Fig 1).

Ninety-four percent of adolescents had high to extremely high excretory salt levels based on the 24-hour urine samples. There was no difference in its distribution in sex, age and socio-economic category (S8 Appendix: S3 Table in S1 File).

The RDA for adolescents specific to their age and sex were adopted from Recommended Dietary Allowance (RDA) for the Indian populations from the Indian Council of Medical Research-National Institute of Nutrition (ICMR-NIN), Hyderabad (2020) [24]. The average daily energy intake was less than the RDA in both sexes. The carbohydrate consumption was above the dietary guidelines. Likewise, protein intake was adequate in both boys and girls. There was gender specific statistically significant difference in protein intake, with boys consuming more. The mean fat intake among adolescents was 47 g/d and it was within RDA for both boys and girls. The fibre intake was lower than the RDA for both boys and girls (Table 1).

Adolescents' overall mean calcium, magnesium and iron intake was less than the RDA. Mean sodium intake was within the RDA of 2000 mg/d. Potassium intake was less than the RDA of ≥3500 mg/d with a mean of 1954 mg/d. Iodine intake assessed from the consumed

**Table 1. Energy and macro-nutrient intake among adolescents.**

| Characteristics | Energy (Kcal/d) | P | Carbohydrates (g/d) | P | Protein (g/d) | P | Total fat (g/d) | P | Fibre (g/d) | P |
|---|---|---|---|---|---|---|---|---|---|---|
| | Mean (SD): (n = 453) | | | | | | | | | |
| Overall | 1813 (594) | | 294 (88) | | 54 (18) | | 47 (26) | | 7 (3) | |
| Sex | | | | | | | | | | |
| Male | 1838 (570) | 0.3 | 297 (83) | 0.4 | 56 (18) | 0.009 | 47 (26) | 0.7 | 8 (3) | 0.05 |
| Female | 1782 (622) | | 290 (94) | | 51 (17) | | 46 (26) | | 7 (3) | |
| Age (years) | | | | | | | | | | |
| 10–12 | 1738 (516) | | 278 (72) | | 52 (19) | | 46 (23) | | 7 (3) | |
| 13–15 | 1829 (602) | 0.5 | 297 (90) | 0.2 | 54 (17) | 0.7 | 47 (26) | 0.9 | 7 (3) | 0.5 |
| 16–18 | 1852 (1109) | | 298 (150) | | 53 (29) | | 50 (46) | | 8 (4) | |

Cut-offs for all macronutrients, minerals and vitamins intake depicted in the figure were adopted from Recommended Dietary Allowance (RDA) for the Indian populations from the Indian Council of Medical Research-National Institute of Nutrition (ICMR-NIN), Hyderabad (2020) [22].

food had a mean of 67 μg/d, less than half of the RDA of ≥150 μg/d. Mean zinc intake was well within the RDA, and mean iodine intake was near the RDA (Table 2).

Vitamin A consumption was lower than the RDA for both sexes. Thiamine and niacin intake were well within the RDA, but riboflavin and vitamin B-6 were slightly below the RDA. Folate consumption was within the RDA for both sexes. However, boys consumed significantly more folate than girls. Consumption of vitamin B-12 was insufficient compared to the RDA. The average vitamin C intake was not as per the RDA among adolescents. It was found that girls consumed more vitamin C (Table 3).

Most adolescents were deficient in the major macro and micronutrients (Fig 2). Carbohydrate was the only macronutrient that was adequate in almost all adolescents. Iodine, fibre, potassium, vitamin B-12 and calcium were the highly deficient nutrients in most adolescents (Fig 2).

## Anthropometric measures

The mean BMI was 17 kg/m$^2$, which was within the normal range compared to the WHO reference curves of adolescents [25]. The waist-hip ratio among boys was 0.85, which was less than the cut-off of 0.9 for males [26]. Similarly, it was 0.83 for girls, which was slightly less than

**Table 2. Mineral intake among adolescents.**

| Characteristics | Calcium (mg/d) | P | Magnesium (mg/d) | P | Iron (mg/d) | P | Sodium (mg/d) | P | Potassium (mg/d) | P | Zinc (mg/d) | P | Iodine (μg/d) | P |
|---|---|---|---|---|---|---|---|---|---|---|---|---|---|---|
| | Mean (SD): (n = 453) | | | | | | | | | | | | | |
| Overall | 626 (257) | | 472 (148) | | 15 (5) | | 1815 (751) | | 1954 (731) | | 9 (3) | | 67 (31) | |
| Sex | | | | | | | | | | | | | | |
| Male | 628 (247) | 0.9 | 488 (154) | 0.01 | 16 (5) | 0.002 | 1860 (776) | 0.2 | 1961 (660) | 0.9 | 9 (3) | 0.009 | 66 (30) | 0.5 |
| Female | 623 (270) | | 453 (137) | | 14 (5) | | 1760 (716) | | 1945 (810) | | 8 (2) | | 68 (32) | |
| Age (years) | | | | | | | | | | | | | | |
| 10–12 | 622 (268) | | 458 (142) | | 14 (5) | | 1693 (788) | | 1865 (627) | | 8 (3) | | 65 (28) | |
| 13–15 | 628 (256) | 0.9 | 475 (148) | 0.6 | 15 (5) | 0.2 | 1838 (739) | 0.2 | 1974 (746) | 0.5 | 9 (3) | 0.7 | 67 (32) | 0.9 |
| 16–18 | 565 (214) | | 462 (243) | | 14 (8) | | 2043 (972) | | 1904 (1149) | | 8 (4) | | 65 (24) | |

Cut-offs for all macronutrients, minerals and vitamins intake depicted in the figure were adopted from Recommended Dietary Allowance (RDA) for the Indian populations from the Indian Council of Medical Research-National Institute of Nutrition (ICMR-NIN), Hyderabad (2020) [22].

**Table 3. Vitamin intake among adolescents.**

| Characteristics | Mean (SD): (n = 453) | | | | | | | | | | | | | | | | |
| --- | --- | --- | --- | --- | --- | --- | --- | --- | --- | --- | --- | --- | --- | --- | --- | --- |
| | Vit A (μg/d) | P | Thiamine (mg/d) | P | Riboflavin (mg/d) | P | Niacin (mg/d) | P | Vit B6 (mg/d) | P | Folate (μg/d) | P | Vit B 12 (μg/d) | p | Vit C (mg/d) | P |
| Overall | 2497 (2526) | | 1.4 (0.5) | | 1.1 (0.4) | | 13 (4.7) | | 1.4 (0.5) | | 248 (106) | | 1.2 (0.8) | | 54 (49) | |
| Sex | | | | | | | | | | | | | | | | |
| Male | 2473 (2671) | 0.8 | 1.5 (0.5) | 0.001 | 1.1 (0.4) | 0.5 | 14 (4.9) | 0.005 | 1.5 (0.5) | 0.02 | 248 (106) | 0.04 | 1.1 (0.8) | 0.6 | 50 (41) | 0.03 |
| Female | 2526 (2345) | | 1.3 (0.5) | | 1.08 (0.4) | | 12 (4) | | 1.4 (0.4) | | 227 (105) | | 1.2 (0.9) | | 59 (57) | |
| Age (years) | | | | | | | | | | | | | | | | |
| 10–12 | 2567 (2978) | | 1.4 (0.5) | | 1.1 (0.4) | | 12.4 (4.7) | | 1.3 (0.4) | | 225 (93) | | 1.2 (0.7) | | 48 (41) | |
| 13–15 | 2459 (2411) | 0.3 | 1.4 (0.5) | 0.5 | 1.1 (0.4) | 0.7 | 13.1 (4.6) | 0.5 | 1.4 (0.5) | 0.4 | 241 (108) | 0.4 | 1.2 (0.8) | 0.8 | 55 (51) | 0.4 |
| 16–18 | 4172 (2980) | | 1.3 (0.8) | | 0.9 (0.5) | | 13.1 (9) | | 1.4 (0.9) | | 249 (119) | | 0.9 (0.3) | | 64 (46) | |

Cut-offs for all macronutrients, minerals and vitamins intake depicted in the figure were adopted from Recommended Dietary Allowance (RDA) for the Indian populations from the Indian Council of Medical Research-National Institute of Nutrition (ICMR-NIN), Hyderabad (2020) [22].

the cut-off of 0.85 for females signifying normal abdominal obesity among boys and a girls (S8 Appendix: S4 Table in S1 File) [26].

Average Mid-Upper Arm Circumference (MUAC) was 21.3 cm. Boys had nearly 2 cm less than the normal range of 23 cm, and girls had 0.5 cm less than the normal range of 22 cm [30].

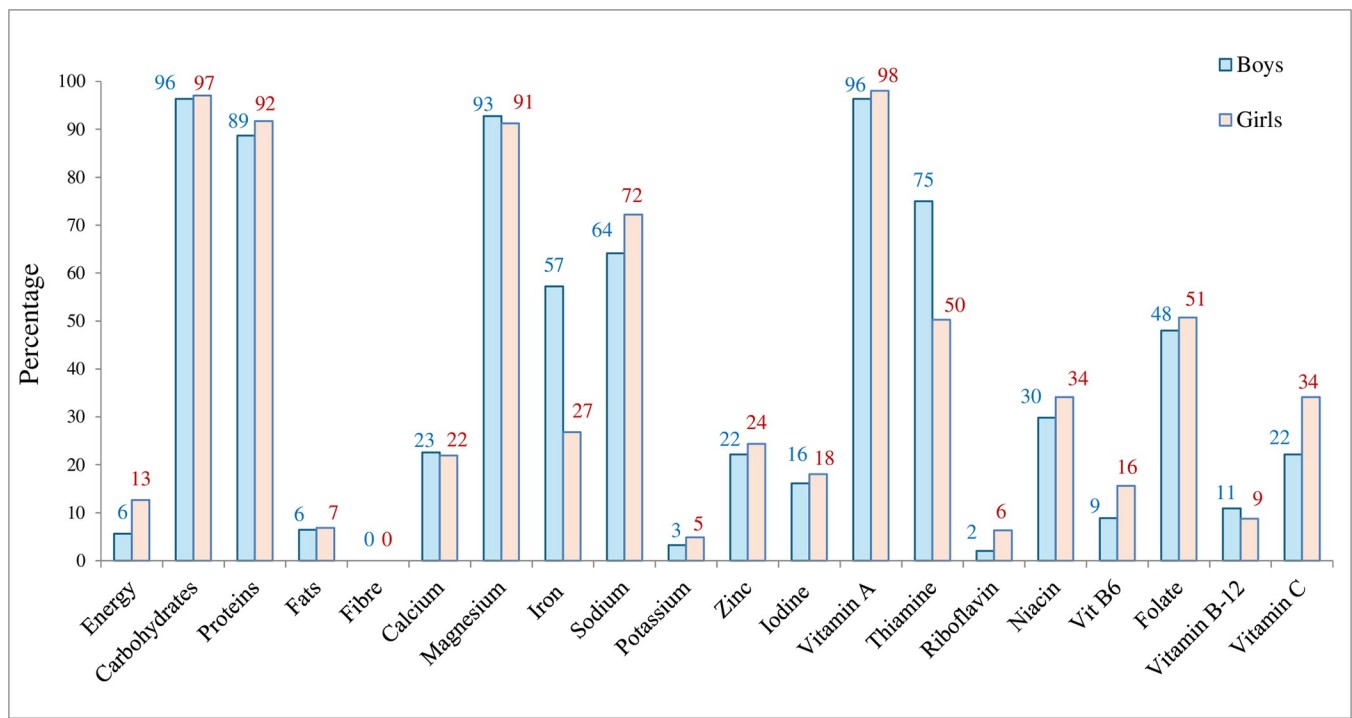

**Fig 2. Proportions of adolescents with adequate intake of nutrients (n = 453).** Cut-offs for all macronutrients, minerals and vitamins intake depicted in the figure were adopted from Recommended Dietary Allowance (RDA) for the Indian populations from the Indian Council of Medical Research-National Institute of Nutrition (ICMR-NIN), Hyderabad (2020) [22], also listed in supplement as S9 Appendix in S1 File.

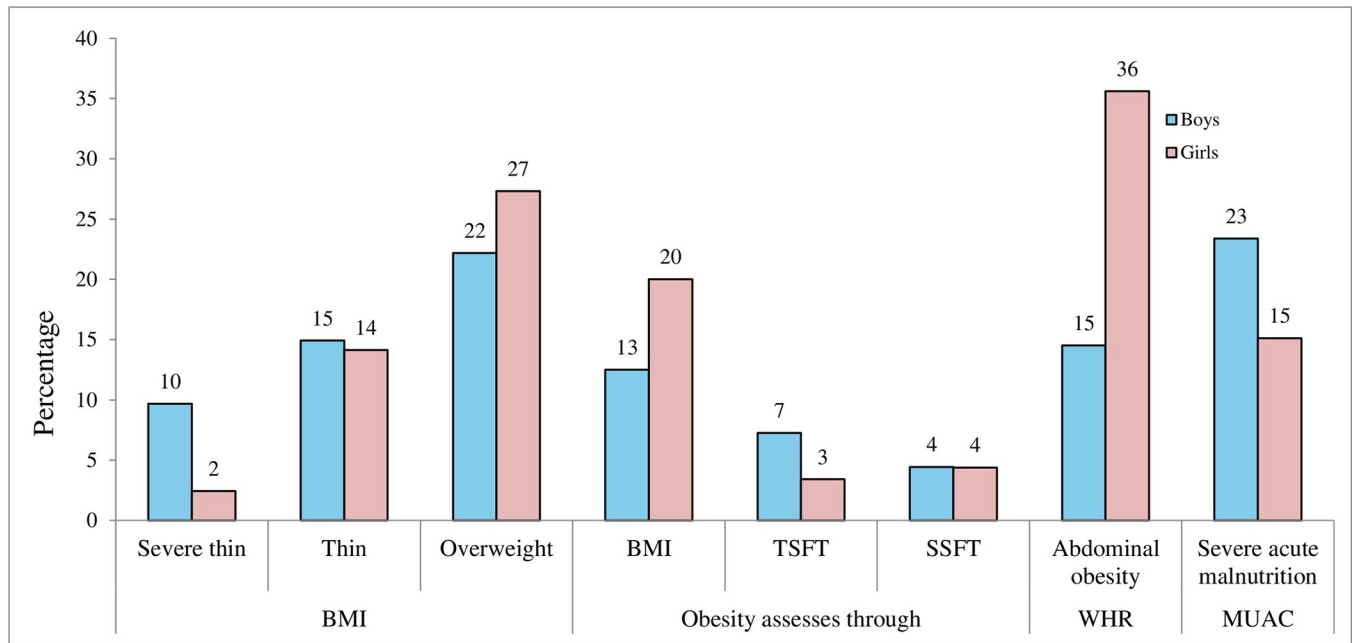

**Fig 3. Malnourishment assessed through anthropometric measures. Cut-offs for malnourishment assessment among adolescents|** Body Mass Index (BMI): adolescents (WHO growth reference percentiles- age and gender specific for 5–19 years, 2007) = ≤ -2SD is severe thinness, = -2SD to -1SD is thinness, -1SD to +1 SD is normal, +1 SD to + 2 SD is overweight and above + 2 SD is obese Triceps skinfold thickness (TSFT) and sub-scapular skinfold thickness (SSFT): Reference curves of Indian adolescents (5–17 years) were used Abdominal Obesity assessed through Waist-Hip Ratio (WHR) = 0.9 for boys and 0.80 for girls Mid-upper Arm Circumference (MUAC) = ≤ 23 cm for boys and ≤22 cm for girls.

It signifies that both the sexes were under-nourished, boys being thinner than the girls. MUAC also has a significant difference among the age groups. The young adolescents had lesser MUAC compared to the older.

Skin-fold thicknesses are a robust method for the assessment of subcutaneous fat. The average TSFT and SSFT were 13.2 mm and 10.7 mm, respectively. These values were in the normal range compared to the reference curves of Indian adolescents (5–17 years) [27]. There was a significant difference among both sexes for the skin folds, with girls having a higher mean for both the TSFT (14.4 mm) and SSFT (12 mm).

Among adolescents, the prevalence of thinness and obesity co-existed. More proportions of boys were thin. Whereas, more proportions of girls were in overweight and obese BMI category. On similar lines, more girls were found to have abdominal obesity. The similar proportion of adolescent boys and girls had high subcutaneous fats assessed through triceps and sub-scapular skinfold thicknesses. A finding in line with another anthropometric measure, MUAC, where higher proportions of boys were found to have severe acute malnutrition (Fig 3).

## Discussion

The nutritional assessment revealed high risk factors for chronic diseases within the study population. Among adolescents in eighth grade, 32% and 55% exceeded the recommended dietary allowances for salt and sugar, respectively, while 90% and 83% exhibited inadequate consumption of fruits and vegetables. Deficiencies in various macro- and micronutrients were prevalent, and a dual burden of overnutrition and undernutrition was observed, with a higher proportion of girls being overweight, having elevated BMI, and abdominal obesity compared to their male counterparts. The mean 24-hour urinary excretion of salt, measured through 24-hour urine samples, was 7 g/d, which is considered the gold standard for assessment.

Adolescents had energy intake below the RDA (S9, S10 Appendices, S5 Table in S1 File). This finding was consistent with other global and Indian studies among adolescents [31, 32]. Boys had a significantly higher intake of proteins than girls, and this finding was consistent with another study among adolescents from other LMICs [33]. Fat intake among adolescents in this study was within the RDA. This finding was in line with another study from India. For girls, the fat intake was slightly higher than the RDA, a result in line with other global studies [33, 34].

This study indicated that intake of iron (15 mg/d), potassium (1954 mg/d), vitamin A (2497 μg/d), and iodine (67 μg/d) was below the RDA. (S9 Appendix: S6 & S7 Tables in S1 File) National Nutrition Monitoring Bureau (NNMB) have also documented that micronutrient intakes were inadequate among Indian adolescents [35]. These findings are comparable to studies from HICs [35, 36]. Generally, insufficient micronutrients, such as iron, have been attributed to a low intake of animal-source foods and potassium and vitamin A to the low consumption of fruits and vegetables. Most of the adolescents from the present study belonged to the low SES groups, where affordability could have been a significant reason for poor micronutrient intakes [36].

The previous studies among adolescents reveal that the mean iodine intake increased by 20% when data was adjusted for the intake of iodised salt [34]. The low level of Iodine intake estimated in the present study can be attributed to the non-inclusion of iodised salt intake in the dietary data analysis. Future studies may include analysis through more accurate measures such as urinary iodine.

The 'BMI for age' is recommended by WHO as the best indicator to assess under-nutrition (thinness) or overweight in adolescents. The present study has adopted the standard methods for determining the nutritional status of adolescents as recommended by the WHO. Various studies across India have adopted the same method for evaluating the nutritional status of adolescents [37].

Underweight (24%) and overweight (10%) co-existed in the present study. Another study from LMIC has reported higher under-nutrition prevalence (45%) [38]. But, the finding was consistent with Indian studies [37, 39]. The prevalence of overweight or obesity in the present study was 12%. Indian studies have reported similar results [37, 39]. The increased prevalence of overweight may be attributed to the increasing accessibility of processed foods and motorised transportation modes. The prevalence of obesity was higher among girls (12%) than boys (9%). This finding is similar to the studies conducted across the country [40, 41]. In contrast, few studies also show a higher prevalence of overweight and obesity among boys when compared to girls [37, 40–42].

Most of the anthropometric indices had higher values among boys. This is supported by the fact that boys usually have a larger body build at this age (10 to 15 years), grow to a larger structure and continue to increase more in height than the girls even after adolescence [42, 43]. Several authors also observed a similar trend from their studies in other states of India and abroad [44]. The difference in the nutritional status assessed through different studies may be due to different cultural environments, the difference in tools used for data collection, and their methodology and analysis. Other classification criteria may also lead to differences in the results. Various classifications such as CDC guidelines 2000, International Obesity Task Force and WHO BMI reference charts etc., are available for estimating nutritional status. Additionally, socio-economic and socio-demographic factors and their associated risk factors may also lead to variation in the results.

The boys' mean WHR was more than that of girls. The mean WHR of the boys and girls were below the WHR ratio cut-offs of 0.9 and 0.85 respectively, signifying normal abdominal obesity. This finding was in line with previous studies [45].

The average TSFT and SSFT were 13.2 mm and 10.7 mm, respectively. The reference curves of Indian adolescents aged (5–17 years) signify that it is in the normal range of subcutaneous fat [27]. Findings were consistent with a study among adolescents in the Netherlands [46]. Average values for both TSFT and SSFT were higher among girls. These findings were consistent with other studies carried out among adolescents of a similar age group in India, Vietnam and Indonesia [47–49]. The findings of 24-hour urinary excretory salt levels from the present study is compared with another study carried out in China among the adolescents [50].

## Methodological considerations, strengths and limitations

The food frequency questionnaire (FFQ) and the 24-hour dietary recall were the commonly used tools in most studies assessing the participants' dietary behaviours as these are quick and inexpensive to administer and have high respondent compliance. From the previous studies, it is evident that FFQ has a higher recall bias compared to the 24-hour dietary recall. The 24-hour dietary recalls have been known to estimate population-level consumption but not accurate individual-level consumption patterns [18]. To overcome this limitation, two 24-hour dietary recalls were recorded on the non-consecutive days for the present study. The two recalls were collected with a time gap of one week to one month. The multiple-pass method was used to collect the dietary behaviours as precise as possible. Standard household measures (e.g. spoons, bowls, glasses and plates) were used to generate actual quantities of ingredients. Another limitation of using 24-dietary recall was underestimating the salt intake [51]. To overcome the limitation of the 24-hour dietary recall tool, the salt intake was also estimated through urinary salt excretion from 24-hour urine samples, a gold-standard method.

Nutritional status is mainly assessed through anthropometric measurements. Different anthropometric measures were used in this study to assess various nutritional aspects of the body, such as height and weight, to assess the individuals' BMI to categorise them into underweight, normal, and overweight categories. For BMI assessment, WHO growth reference charts for 5 to 19 years are not specific for Indian Adolescents, yet they are widely used [25]. BMI does not directly measure body fat and does not distinguish between muscle and body fat weight. BMI may overestimate the body fat in a person with a muscular build and may underestimate body fat in older adults who have lost muscle. Therefore, it was considered best to use BMI with other measures, such as WHR and MUAC [52].

Waist and hip circumference are used to assess WHR, providing information regarding abdominal obesity [52]. MUAC is the body measurement least affected by weight and height changes compared to anthropometric measures such as BMI and WHR [53]. Therefore, MUAC was recorded to assess acute malnutrition among adolescents. However, MUAC fails to account for sex and age-specific differences as it is generally estimated through a single cut-off value [54]. The MUAC among the adolescents from the present study was on the lower end, signifying under-nutrition. The adolescents from the present study were from public schools, which usually cater to the lower socio-economic strata of the population, where the proportion of starvation among adolescents is higher than the other SES groups. However, there is a lack of evidence-based MUAC cut-offs internationally. Due to this gap, several countries have established cut-offs that vary widely. Additionally, various skin fold thicknesses such as TSFT and SSFT were used to assess the subcutaneous fats.

Before the final data collection, pilot testing was undertaken to assess the protocol's feasibility and the adequacy of the research instruments [55]. The study's strength was using validated and pretested tools in the Indian scenario and methodological data collection guidelines for the nutritional assessment.

Only public schools were enrolled in the study, which usually cater to the lower socio-economic groups in India. The implication of excluding private schools might lead to a lack of representation of participants across various socio-economic groups, limiting generalizability. However, the low-middle income group is a significant section of Indian society, and their representation in the study sample is one of the strengths of this study.

## Conclusions

Adolescents from public schools of UT Chandigarh had poor nutritional status, with inadequate energy, fibre, iron, potassium, iodine, calcium, folate, vitamin A, vitamin B-12 and vitamin C intakes.

Schools should regularly check the nutritional status of adolescents and may use the tools and measures used in the present study. However, in case of time constraints and lack of resources, schools may use only BMI or waist circumference as assessment criteria for regular monitoring of nutrition status among adolescents. Those identified as an underweight category through BMI may be assessed for their MUAC to know the severity of the undernourishment. In comparison, those falling in the overweight or obese categories of BMI may be assessed for WHR to determine central obesity among adolescents. Mid-day meals should be regularly checked for their nutritional values and provided per the age requirements of the adolescents.

There is a need for behaviour change interventions to improve nutritional status in early life. Parents' and teachers' involvement can have far-reaching benefits regarding the sustainability of behavioural change for healthy dietary behaviours. The school-based intervention activities to improve nutrition among adolescents should be developed and implemented in a manner that can be replicated in other settings in India and low-income countries with modifications to account for differences in the contextual factors. The degree and type of activities must depend on the target populations' social and economic status and cultural aspects at a given time.

## Supporting information

**S1 Checklist. Ensures adherence to reporting guidelines for observational studies.**
(PDF)

**S1 File. Includes all appendices referenced in the manuscript.**
(PDF)

**S1 Protocol. Detailed documentation of the study methodology.**
(PDF)

## Acknowledgments

We extend our heartfelt gratitude to the George Institute of Global Health, Hyderabad, for facilitating the laboratory testing of urine samples collected for this study.

## Author Contributions

**Conceptualization:** Sandeep Kaur, Rajesh Kumar, Manmeet Kaur.

**Data curation:** Sandeep Kaur.

**Formal analysis:** Sandeep Kaur.

**Funding acquisition:** Manmeet Kaur.

**Investigation:** Sandeep Kaur.

**Methodology:** Sandeep Kaur, Rajesh Kumar, Manmeet Kaur.

**Project administration:** Sandeep Kaur.

**Resources:** Rajesh Kumar, Manmeet Kaur.

**Software:** Rajesh Kumar.

**Supervision:** Rajesh Kumar, Manmeet Kaur.

**Validation:** Rajesh Kumar, Manmeet Kaur.

**Writing – original draft:** Sandeep Kaur, Rajesh Kumar, Manmeet Kaur.

**Writing – review & editing:** Sandeep Kaur, Rajesh Kumar, Manmeet Kaur.

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
