## [Decision Letter · Decision Letter 0]

22 Oct 2023

PONE-D-23-18349Nutritional Assessment of Adolescents: a Cross-sectional Study from Public Schools of North IndiaPLOS ONE

Dear Dr. Kaur,

Thank you for submitting your manuscript to PLOS ONE. After careful consideration, we feel that it has merit but does not fully meet PLOS ONE’s publication criteria as it currently stands. Therefore, we invite you to submit a revised version of the manuscript that addresses the points raised during the review process.

We look forward to receiving your revised manuscript.

Kind regards,

George Kuryan

Academic Editor

PLOS ONE

Journal Requirements:

Reviewers' comments:

Reviewer's Responses to Questions

**Comments to the Author**

1. Is the manuscript technically sound, and do the data support the conclusions?

Reviewer #1: Partly

2. Has the statistical analysis been performed appropriately and rigorously? 

Reviewer #1: No

3. Have the authors made all data underlying the findings in their manuscript fully available?

Reviewer #1: No

4. Is the manuscript presented in an intelligible fashion and written in standard English?

Reviewer #1: Yes

5. Review Comments to the Author

Reviewer #1: i have made my comments in the manuscript attached with my comments.

authors need the assistance of a statistician. Many statistical/Analyses errors. reference standards are not correctly used for MUAC. Age range of participants is 10-16 with different cutoffs for MUAC. Cannot take average.

Tables are not adequately labelled.

whether the values are mean, median with SD/IQR not labelled in the tables

Reference standards for macronutrients and micronutrients are not mentioned

6. PLOS authors have the option to publish the peer review history of their article (what does this mean?). If published, this will include your full peer review and any attached files.

Reviewer #1: No

---

## [Author Response · Author response to Decision Letter 0]

19 Jan 2024

1. In this result section, need not write the need of health promotion interventions

 Thanks for your input. As suggested, we have removed this statement.

2. Mention the % of energy intake from carbohydrate referring to dietary goals . Include in table 2 the reference standards-dietary goal % 

 Thank you. Based on your suggestion, we have included all the reference standards used for macro and micronutrients in the 

 supplement as appendix S9. We have also included these in the text in the lines 207-215.

3. Please include all RDA in the table 2 in brackets 

 We thank the reviewer for this valuable suggestion. However, as the table was too difficult to comprehend with the addition of RDA 

 values in the brackets, we have provided RDA in the supplement as separate tables in Appendix S9. 

4. To state adolescents are deficient in macro and micronutrients , RDA should be mentioned in the text and table

 We agree and have provided these details in the manuscript as figure 2, as tables in tables S4, S5 and S6, and at various places in the 

 text from lines 189-249.

5. Create 95% confidence interval to see whether the cut-off is included within the CI to say it is normal or less 

 Thank you for highlighting this point. Based on the comments, we have added the 95% CI in lines 251-255. 

6. What does the authors mean by normal abdominal obesity?

 We thank the reviewer for providing an opportunity for clarification. With 'normal abdominal obesity, ' authors meant the waist-hip ratio 

 that falls within the normal range, which is the waist-hip ratio of ≤ 0.9 for males and ≤ 0.85 for females. However, we agree that the 

 term may confuse the readers. Therefore, we have changed it to 'no abdominal obesity'.

7. Optimal MUAC cut off point is different for different age groups in your study (10-16 yrs) . Cannot take an average 

 Thank you! We want to clarify that mid- upper arm circumference (MUAC) depicts severe malnourishment. We agree that MUAC 

 measurements can vary widely for adolescents based on factors like growth spurts, gender, and individual body composition. However, 

 there are no universally accepted cut-offs in this age group. Moreover, a single cut-off point of 23 cm for males and 22 cm for females is 

 widely used. We have added this limitation of MUAC in the discussion from lines 360-370, supported by reference 52 in the manuscript. 

8. Clearly mention in this table what each value represents. What is 88? Is it SD? What is 294? Is it mean or median? Label it. 

 Thanks to the reviewer, we would like to clarify that we have added these details at the top of the table. The values provided are in 

 Mean (SD).

---

## [Decision Letter · Decision Letter 1]

4 Jun 2024

PONE-D-23-18349R1Nutritional Assessment of Adolescents: a Cross-sectional Study from Public Schools of North IndiaPLOS ONE

Dear Dr. Kaur,

Thank you for submitting your manuscript to PLOS ONE. After careful consideration, we feel that it has merit but does not fully meet PLOS ONE’s publication criteria as it currently stands. Therefore, we invite you to submit a revised version of the manuscript that addresses the points raised during the review process.

We look forward to receiving your revised manuscript.

Kind regards,

George Kuryan

Academic Editor

PLOS ONE

Reviewers' comments:

Reviewer's Responses to Questions

**Comments to the Author**

1. If the authors have adequately addressed your comments raised in a previous round of review and you feel that this manuscript is now acceptable for publication, you may indicate that here to bypass the “Comments to the Author” section, enter your conflict of interest statement in the “Confidential to Editor” section, and submit your "Accept" recommendation.

Reviewer #2: (No Response)

2. Is the manuscript technically sound, and do the data support the conclusions?

Reviewer #2: No

3. Has the statistical analysis been performed appropriately and rigorously? 

Reviewer #2: No

4. Have the authors made all data underlying the findings in their manuscript fully available?

Reviewer #2: No

5. Is the manuscript presented in an intelligible fashion and written in standard English?

Reviewer #2: Yes

6. Review Comments to the Author

Reviewer #2: Abstract:

Needs to be abridged and made crisp after addition of more information. It needs to add definitions/cut-offs for the various parameters used. How were the dietary history converted to micronutrient intakes etc. Might want to restrict to key parameters. Not sure why it is recommended that schools should regularly assess adolescent nutritional status. I am sure it will also apply for children. So are we saying that all schools should have this process as a part of regular work/duties and what kind of intervention is to be done by them. Or this is a part of the school health program of the health system.

Main Paper:

Introduction: The last line should read public schools in a city in North India.

A national level adolescent survey for NCD risk factors has been conducted as a part of the NNMS, which has not been referred to at all. Why? Mathur P. etal. Baseline risk factor prevalence among adolescents aged 15-17 years old: findings from NNMS of India. BMJ Open. 2021 Jun 29;11(6):e044066.

Methods:

….. 106 public schools had 8th grade classes and “on average 10790 adolescents study in the 8th grade. Not clear – average per school or is it total and not average?

Sampling of schools and classes and adolescents is not well described. If the sample size was 360 , why were 462 sampled?

Not clear why is analysis of school curriculum is referred to?

Primary outcome variable has been labeled as Nutritional status. It is not clear what does this status refer to (anthropometry, dietary intake?)

Urinary salt excretion is a reasonably good measure of population level sodium intake and not at individual level. A single measurement is not appropriate for estimation of salt intake at individual level. Details of sodium assessment including the formula used should be provided. Reference 27 seems inappropriate to define cut-off at individual level for sodium intake.

PURE study software should have a reference.

Reference to senior faculty member and DC members should be deleted.

The cut-offs for Waist circumference and MUAC seem to be for adults. Are these adolescent-specific?

The need for p-values is not justified given that there were no hypotheses to begin with.

Results:

Table 1 can be deleted and matter in the text is sufficient. At best can be kept as a supplementary table.

While in the results fat intake is classified as slightly below RDA 9provides 25% of energy) in the Figure 2, only 6-7% have adequate intake? Seems to be a discrepancy.

I have reservations about the use of theoretical levels of micronutrients to estimate their intake as we know these vary based on the soil composition and manure. There is too much emphasis on micronutrients in the paper. In the absence of biochemical estimation even in a subsample, I would not focus on them.

Figure 1 The use of Nar chart is inappropriate as it totals to 100%. Pie or a stacked bar chart is a better alternative.

Figure 3. it says severe acute malnutrition by MUAC and the cut off in legend is MUAC> 23/22 cms!!

7. PLOS authors have the option to publish the peer review history of their article (what does this mean?). If published, this will include your full peer review and any attached files.

Reviewer #2: No

---

## [Author Response · Author response to Decision Letter 1]

12 Jul 2024

RESPONSE TO REVIEWER'S COMMENTS

Response to Reviewer 2

We thank the reviewer for the comments, which helped us improve the paper. Further, in light of valuable comments from the reviewer, we have revised the manuscript by addressing all comments. The response to each comment is provided in blue, and the manuscript changes are highlighted using the 'track changes' command. '

1. An abstract needs to be abridged and made crisp after addition of more information. It needs to add definitions/cut-offs for the various parameters used. How were the dietary history converted to micronutrient intakes etc. Might want to restrict to key parameters. 

We thank the reviewer for the input. Based on this we have added more details in the abstract. 

2. Not sure why it is recommended that schools should regularly assess adolescent nutritional status. I am sure it will also apply for children. So are we saying that all schools should have this process as a part of regular work/duties and what kind of intervention is to be done by them. Or this is a part of the school health program of the health system.

We appreciate the reviewer providing us with the opportunity to clarify this important aspect of our study. The literature indicates that behavioural risk factors, such as unbalanced diets, begin to emerge in early adolescence. These risk factors are known to lead to chronic diseases later in life. In low- and middle-income countries like India, there is a double burden of malnutrition, with a significant proportion of adolescents being either underweight or overweight.

Schools are critical environments where children spend most of their day and learn from their teachers and peers. Therefore, schools can be ideal settings to inculcate healthy behaviours early in life. Currently, in schools, there is only one chapter in the sixth-grade textbook covering the sources, components, and importance of nutrition. However, there is no specific chapter on this crucial aspect of health until the tenth grade.

Our study was part of a larger Cluster Randomized Trial (referred to in the manuscript as number 11) aimed at assessing the impact of a health promotion intervention package on reducing behavioural risk factors related to chronic diseases. One component of the intervention focused on dietary behavioural risk factors, such as high salt and sugar intake and low fruit and vegetable intake.

Despite the challenges posed by the COVID-19 pandemic, our study was effective in reducing some of these behavioural risk factors. We propose that future studies build on this work to develop a module that can be easily integrated into the school curriculum. This would promote healthy behaviours from early life, potentially reducing the ever-increasing burden of chronic diseases in the future.

3. Introduction: The last line should read public schools in a city in North India.

Thank you, we have corrected the sentence as suggested. 

4. A national level adolescent survey for NCD risk factors has been conducted as a part of the NNMS, which has not been referred to at all. Why? Mathur P. etal. Baseline risk factor prevalence among adolescents aged 15-17 years old: findings from NNMS of India. BMJ Open. 2021 Jun 29;11(6):e044066.

We thank the reviewer for bringing this to our attention. While we did refer to NNMS (reference number 4) in our manuscript, we have now included the additional reference (number 5) based on the reviewer’s suggestion.

5. Methods:….. 106 public schools had 8th grade classes and “on average 10790 adolescents study in the 8th grade. Not clear – average per school or is it total and not average?

We would like to clarify that data collected from the Department of School Education Chandigarh revealed that 106 public schools had eighth-grade classes, and on average 10,790 adolescents study in 8th grade in these schools. Most schools had four to five sections in 8th grade class. Each section of the class had about 30 adolescents.

6. Sampling of schools and classes and adolescents is not well described. If the sample size was 360 , why were 462 sampled?

We thank the reviewer for providing us with the opportunity to address this crucial aspect of our study. Literature suggests that the behavioural risk factors of chronic diseases start to emerge during early adolescence. Therefore, based on this knowledge and analysis of the school curriculum in UT Chandigarh, adolescents from 8th grade studying in public schools were selected for this study. 

A total of twelve clusters (schools) were sampled based on the formula mentioned below. 

c = 1 + f [ π0 (1 - π0)/m + π1(1 - π1)/m + k2 (π0 2 + π1 2)] / (π0 - π1)2 

Where c is the number of clusters required, f = 7.84 for 5% type I error and 80% power, π1 is the expected proportion of the behavioural risk factor in the intervention arm, π0 is the expected proportion of risk factor in the comparison arm, m is the number of individuals in each cluster (assumed equal in all clusters), and k is the coefficient of variation in the (true) proportions between clusters in each study arm. 

Data collected from the Department of School Education Chandigarh revealed that 106 public schools had eighth-grade classes, and on average 10,790 adolescents study in 8th grade in these schools. Most schools had four to five sections in 8th grade class. Each section of the class had about 30 adolescents. Therefore, the cluster size (m) was 30 adolescents in each section. 

The c value for each primary outcome was calculated separately using data for adolescents from recent studies. Out of all risk factors, the maximum c value (12) was considered the sample size of the number of clusters for the study, i.e., sugar intake. Thus, a sample size of 360 adolescents (12 clusters x 30 adolescents per cluster) was estimated to provide valid statistical inference about the impact of the intervention.

However, in all twelve consenting schools, the eligible eighth-grade adolescents were 462. All were included in the study owing to ethical considerations. The details of the sampling and sample size are provided in the protocol paper of the study which has been referred to in the manuscript. 

7. Not clear why is analysis of school curriculum is referred to?

Analysis of school curriculum was referred to as the present study was part of a larger cluster randomized controlled trial to assess the impact of health promotion intervention. It was important to incorporate a health promotion intervention package in the school settings without disturbing the school curriculum for easy replication of the study for sustainable behaviours.

8. Primary outcome variable has been labeled as Nutritional status. It is not clear what does this status refer to (anthropometry, dietary intake?)

Thank you! We have corrected it to ‘Dietary Intake’. 

9. Urinary salt excretion is a reasonably good measure of population level sodium intake and not at individual level. A single measurement is not appropriate for estimation of salt intake at individual level. Details of sodium assessment including the formula used should be provided. 

We acknowledge the reviewer's point regarding urinary salt excretion as a measure more suited for assessing population-level sodium intake rather than individual-level intake. However, it's important to note that urinary sodium excretion is widely regarded as the 'gold-standard' method for analyzing sodium levels. Our decision to include this measure in the study alongside dietary history was aimed at mitigating the limitations of 24-hour dietary recalls, which tend to underestimate salt levels.

Moreover, urinary excretion provides an objective parameter that necessitates laboratory testing, thus enhancing the accuracy of assessing salt levels. 

10. Reference 27 seems inappropriate to define cut-off at individual level for sodium intake.

Thank you, the paper is a systematic review which provided cut-offs from urinary sodium at 3.3.4. section of the referred paper. 

11. PURE study software should have a reference.

Added it as a reference in the manuscript with number 20. 

12. Reference to senior faculty member and DC members should be deleted.

Thank you for the input. We have deleted it. 

13. The cut-offs for Waist circumference and MUAC seem to be for adults. Are these adolescent-specific?

Waist-circumference and MUAC have single-point cut-offs. These are for adolescents. The reference for this has been provided as numbers 26 and 27. 

14. The need for p-values is not justified given that there were no hypotheses to begin with.

Thank you, we have added a hypothesis for the study in the methods. 

15. Results:Table 1 can be deleted and matter in the text is sufficient. At best can be kept as a supplementary table.

Thank you. Based on the reviewer’s suggestion we have shifted Table 1 to the supplement as Appendix 8: Table S1. 

16. While in the results fat intake is classified as slightly below RDA (provides 25% of energy) in the Figure 2, only 6-7% have adequate intake? Seems to be a discrepancy.

Thank you for your question. However, we'd like to clarify that while the fat intake may be slightly below the Recommended Dietary Allowance (RDA), it exhibits a consistent pattern across all age and sex categories. Upon re-evaluating the results, we found no discrepancy, with only 6% of boys and 7% of girls demonstrating adequate fat intake.

17. I have reservations about the use of theoretical levels of micronutrients to estimate their intake as we know these vary based on the soil composition and manure. There is too much emphasis on micronutrients in the paper. In the absence of biochemical estimation even in a subsample, I would not focus on them.

We appreciate the input from the reviewer and would like to clarify our methodology for estimating micronutrients. To estimate both micro and macronutrients, such as protein, carbohydrates, fat, vitamins, and minerals, we utilized the PURE study computer software. This software has been specifically developed for assessing the intake of Indian foods and has previously been validated in Indian settings.

The tool encompasses a comprehensive list of 143 different food items commonly consumed in India. Upon entering a participant's average portion size and frequency of food consumption per day into the software, it can estimate 72 different macro and micronutrients for that individual.

Our approach involved inputting participants' dietary data, including portion sizes and consumption frequency, into the software. Based on the Indian dietary data provided by the ICMR-NIN (Indian Council of Medical Research-National Institute of Nutrition), the software employs the weight of raw ingredients in the food items. The NIN furnishes nutrient data per 100 grams of raw food ingredients. Therefore, the software utilizes this data in proportion to the quantity of ingredients used to prepare the food items.

In our methodology, the software estimates the quantity of nutrients based on the raw ingredients typically used in food preparation. This allows for the estimation of various macro and micronutrients present in the raw ingredients of the consumed food items.

While dietary assessment provides valuable insights, we agree that it may not be as accurate as biochemical assessment for micronutrient evaluation. For additional clarity on our methodology, please refer to Appendix S3 and S4 included in the supplement.

Some of the earlier published studies have used the same dietary tool to assess micro and macronutrients. 

1. Kaur J, Kaur M, Chakrapani V, Webster J, Santos JA, Kumar R. Effectiveness of information technology-enabled 'SMART Eating' health promotion intervention: A cluster randomized controlled trial. PLoS One. 2020 Jan 10;15(1):e0225892. doi: 10.1371/journal.pone.0225892.

2. Mahajan R, Malik M, Bharathi AV, Lakshmi PV, Patro BK, Rana SK, Kumar R. Reproducibility and validity of a quantitative food frequency questionnaire in an urban and rural area of northern India. Natl Med J India. 2013 Sep-Oct;26(5):266-72.

18. Figure 1 The use of Nar chart is inappropriate as it totals to 100%. Pie or a stacked bar chart is a better alternative.

We express our gratitude to the reviewer for their valuable suggestion. Following their advice, we have made the necessary adjustments to the figure, now presenting it in a stacked bar format.

19. Figure 3. it says severe acute malnutrition by MUAC and the cut off in legend is MUAC> 23/22 cms!!

Yes, the figure shows severe acute malnutrition by MUAC separately for boys and girls. The cut-off point of MUAC for adolescent boys is ≥ 23cm and for girls is ≥ 22cm. So, the figure depicts that out of all boys recruited 23% were severely malnourished, i.e., 23% of boys had MUAC equal to or below 23 cm and similarly 15% of girls had MUAC equal to or below 22cm.

---

## [Decision Letter · Decision Letter 2]

27 Nov 2024

PONE-D-23-18349R2Nutritional Assessment of Adolescents: a Cross-sectional Study from Public Schools of North IndiaPLOS ONE

Dear Dr. Kaur,

Thank you for submitting your manuscript to PLOS ONE. After careful consideration, we feel that it has merit but does not fully meet PLOS ONE’s publication criteria as it currently stands. Therefore, we invite you to submit a revised version of the manuscript that addresses the points raised during the review process.

Please submit your revised manuscript by Jan 11 2025 11:59PM. you will need more time than this to complete your revisions, please reply to this message or contact the journal office at plosone@plos.org. Please include the following items when submitting your revised manuscript:

We look forward to receiving your revised manuscript.

Kind regards,

George Kuryan

Academic Editor

PLOS ONE

Journal Requirements:

Reviewers' comments:

Reviewer's Responses to Questions

**Comments to the Author**

1. If the authors have adequately addressed your comments raised in a previous round of review and you feel that this manuscript is now acceptable for publication, you may indicate that here to bypass the “Comments to the Author” section, enter your conflict of interest statement in the “Confidential to Editor” section, and submit your "Accept" recommendation.

Reviewer #3: All comments have been addressed

Reviewer #4: (No Response)

2. Is the manuscript technically sound, and do the data support the conclusions?

Reviewer #3: Partly

Reviewer #4: Yes

3. Has the statistical analysis been performed appropriately and rigorously? 

Reviewer #3: I Don't Know

Reviewer #4: Yes

4. Have the authors made all data underlying the findings in their manuscript fully available?

Reviewer #3: Yes

Reviewer #4: Yes

5. Is the manuscript presented in an intelligible fashion and written in standard English?

Reviewer #3: Yes

Reviewer #4: Yes

6. Review Comments to the Author

Reviewer #3: This is a very relevant study and has the potential to provide useful information

1. Line 173 can be re-structured to make it more comprehensible (32% and 55% of salt and sugar line)

2. MUAC is an unusual metric to assess adolescents because of variable growth spurts - it as shown an unusual proportion of 'Severe acute malnutrition' which is not corroborated by the BMI values.

3. Reference 30 for MUAC does not state that less than 22/23 is malnutrition and definitely not severe malnutrition. SO figure 3 is a misrepresentation

4. Lines 326 to 335 seem redundant. If the authors say that MUAC is not ideal and also state that because it is a public school and the students are anyways starving (line 331) then that makes the null hypothesis irrelevant. SO the basic premise of the study is debunked - suggesting that this is an ad hoc analysis of existing data.

5. The conclusions and recommendations do not align with the results. This was not a health systems/policy research/analysis to suggest systemic changes. This is an observational study which can provide suggestions for further studies.'

6. The p values mentioned in the tables - what statistical tests were used? Were non parametric tests used for relevant variables?

Reviewer #4: Thank you for the opportunity. The study provides a good insight into nutrition among adolescents attending public schools in north India. I have a few questions and suggestions.

1. The introduction, methodology and discussion can be more concise. For example, the PURE software is explained in both the data collection and data management sections. There is no need to include investigator initials in the methodology and should be restricted to the author contribution section. A large part of the discussion seems to be reiterating results. Rather, the implications of the findings, and comparisons should be on focus. The discussion seems to focus on the proportion of risk factors while the results section mostly provides the mean and standard deviation. The strengths and limitations section does not mention specific strengths or problems identified in pilot testing. Some limitations seem to be mentioned before this section starts. This should be organized better.

2. It is my opinion that an initial survey of an CRCT is exploratory and does not need a separate hypothesis (especially not a post hoc one). Further, the post hoc hypothesis added in this revision is ineffectual since it is not related to the comparison between the groups.

3. The methodology section mentions chi square, t-test and ANOVA. These are parametric test and assume normality. Has the distribution been checked? Since the authors mention most students belong to the lower SES the results may also be skewed.

4. There are some findings that seem inconsistent. The discussion mentions that the prevalence of overweight and obesity was 12% among girls and 9% in boys (Line 278), whereas Figure 3 has overweight at 27 and 22% respectively. Further according to Figure 2 only 13% and 6% of girls and boys respectively have adequate energy intake which is not consistent with finding of overweight and obesity. The age distribution mentioned 10-16 is quite broad for a single grade. Is this expected and accurate? Does this impact the results?

5. The manuscript needs a full revision and copy editing to weed out grammatical and typographical errors. I've mentioned a few examples below

a. Please consider the statement " Adolescents were deficient in most macro and micro-nutrients, such as energy, fats, fibre, iron, zinc, iodine, riboflavin, and vitamins B-6 and B-12" within the results section in the abstract. Is energy considered a macronutrient? If not, the sentence should be rephrased.

b. Some sentences have confusing syntax and need to be rephrased: Example line 29 "Owing to expensive, lifelong treatments and loss of productive life years due to the increasing premature deaths and disabilities related to various chronic diseases in India, it is recommended to focus more on promotion and prevention than curative measures."

c. The term "normal abdominal obesity" has been used multiple times in the manuscript. Normal obesity is an oxymoron. please use a more accurate term.

d. Line 292 on Page 14. "The mean WHR of the boys and girls were below the WHR ratio cut-offs of 0.85 and 0.9 respectively....". When using the term "respectively" the two sets of nouns should be in the same order. This sentence implies that the cutoff for boys is 0.85 and for girls 0.9.

e. Figure 3 says severe acute malnutrition by MUAC and the cut off in legend is MUAC≥ 23/22 cms. The symbol "≥" connotes "greater than" and is inaccurately used here.

7. PLOS authors have the option to publish the peer review history of their article (what does this mean?). If published, this will include your full peer review and any attached files.

Reviewer #3: **Yes: **Jackwin Sam Paul G

Reviewer #4: No

---

## [Author Response · Author response to Decision Letter 2]

10 Dec 2024

Reviewer 3

It is a very relevant study and has the potential to provide useful information

We thank the reviewer for the comments, which helped us improve the paper. Further, in light of valuable comments from the reviewer, we have revised the manuscript by addressing all comments. The response to each comment is provided in blue, and the manuscript changes are highlighted using the 'track changes' command. 

1. Line 173 can be re-structured to make it more comprehensible (32% and 55% of salt and sugar line)

Thank you! Based on the reviewer’s suggestion, we have revised the line to make it more concise and clear. 

2. MUAC is an unusual metric to assess adolescents because of variable growth spurts - it as shown an unusual proportion of 'Severe acute malnutrition' which is not corroborated by the BMI values.

We would like to thank the reviewer for providing the opportunity to clarify this important point. As highlighted in the discussion section, MUAC is the anthropometric measure least affected by changes in weight and height, compared to other measures such as BMI and Waist-Hip Ratio (WHR). For this reason, MUAC was selected to assess acute malnutrition among adolescents, in addition to BMI, which is more sensitive to the fluctuations in weight and height that are common during adolescence.

3. Reference 30 for MUAC does not state that less than 22/23 is malnutrition and definitely not severe malnutrition. SO figure 3 is a misrepresentation

We would like to express our gratitude to the reviewer for their valuable feedback and take this opportunity to clarify a few points regarding reference 30. In Section 3.1.1 of this reference, a MUAC cut-off of 22 cm for adolescents is provided. Additionally, Figure 2 in the same reference summarizes the MUAC cut-offs for adolescents, adults, and pregnant women, based on existing literature. As mentioned in lines 332-334 of the discussion section, there is a recognized lack of internationally established, evidence-based MUAC cut-offs, which has resulted in variations in the thresholds used across different countries. In light of this gap, we have adopted the MUAC cut-off presented in reference 30 for the current study. Furthermore, we would like to clarify that MUAC, as a single cut-off measure, is commonly used to assess severe acute malnutrition.

4. Lines 326 to 335 seem redundant. If the authors say that MUAC is not ideal and also state that because it is a public school and the students are anyways starving (line 331) then that makes the null hypothesis irrelevant. SO the basic premise of the study is debunked - suggesting that this is an ad hoc analysis of existing data.

Thank you! We would like to take this opportunity to clarify that in lines 326-335, we are conveying that MUAC was used alongside other assessment measures, such as BMI and WHR. MUAC is less affected by changes in weight and height, which is particularly important during adolescence, a growth phase where BMI and WHR alone may not fully capture nutritional status. We aimed for a more holistic approach by incorporating MUAC, a measure that we believe is robust despite its limitations. We did not state that MUAC is not ideal; rather, we acknowledged its potential limitations, including its use as a single cut-off measure without a universally accepted threshold.

These points in the manuscript are discussed in the context of our findings, where we report that 23% of boys and 15% of girls in our study were severely malnourished. We attribute these high figures to the fact that our participants were from public schools, which primarily serve lower socioeconomic groups, where access to nutritious food is often limited. 

5. The conclusions and recommendations do not align with the results. This was not a health systems/policy research/analysis to suggest systemic changes. This is an observational study which can provide suggestions for further studies.'

We appreciate the reviewer’s feedback. We acknowledge that our study is an observational study rather than a health systems or policy analysis. The conclusions and recommendations provided were intended to highlight key findings from the study and suggest areas for further research. Our conclusions emphasise the need for further and regular investigations into the nutritional status of adolescents. Given that our study reveals a high burden of malnutrition among adolescents, we also recommend that future research focus on interventions aimed at promoting healthy dietary behaviours to address this ongoing issue. Howeever, based on the reviewer’s comment we have revised the section to ensure that the recommendations are intended to guide future research in this area.

6. The p values mentioned in the tables - what statistical tests were used? Were non parametric tests used for relevant variables?

Thank you! In our study, the data were normally distributed, and therefore, parametric tests such as the t-test and ANOVA were used to calculate the p-values reported in the tables. No non-parametric tests were applied, as the assumptions of normality were met for the relevant variables. We have ensured that the statistical tests used are clearly outlined in the lines 134-143. 

Reviewer 4: 

Thank you for the opportunity. The study provides a good insight into nutrition among adolescents attending public schools in north India. 

We thank the reviewer for their time to review the manuscript. We agree to all the issues that have been raised. The response to each of the comments is provided in blue colour. The changes have been made in the manuscript using the ‘track changes’ command. 

1. The introduction, methodology and discussion can be more concise. For example, the PURE software is explained in both the data collection and data management sections. There is no need to include investigator initials in the methodology and should be restricted to the author contribution section. A large part of the discussion seems to be reiterating results. Rather, the implications of the findings, and comparisons should be on focus. The discussion seems to focus on the proportion of risk factors while the results section mostly provides the mean and standard deviation. The strengths and limitations section does not mention specific strengths or problems identified in pilot testing. Some limitations seem to be mentioned before this section starts. This should be organized better.

We thank the reviewer for their valuable input. In response, we have revised the introduction, methods, and discussion sections to make them more concise, incorporating the suggestions provided. The discussion section has been further refined to emphasise the implications of the findings and facilitate comparisons. Additionally, we have reorganised these sections to improve the overall flow. We would also like to clarify that the proportions mentioned in the discussion section are derived from Figure 1. Finally, the strengths and limitations have been restructured for better coherence.

2. It is my opinion that an initial survey of an CRCT is exploratory and does not need a separate hypothesis (especially not a post hoc one). Further, the post hoc hypothesis added in this revision is ineffectual since it is not related to the comparison between the groups.

Thank you for your valuable feedback. However, we would like to clarify that while the baseline survey for the CRCT was used to inform the intervention, it provides important data that reflects the ground reality and we think it should be shared. We also wish to highlight that participant allocation to the intervention and the comparison arms was only done after the completion of the baseline survey. Therefore, for this CRCT, the baseline survey functions as a cross-sectional survey rather than just an exploratory one. 

3. The methodology section mentions chi square, t-test and ANOVA. These are parametric test and assume normality. Has the distribution been checked? Since the authors mention most students belong to the lower SES the results may also be skewed.

We thank the reviewer for providing us with the opportunity to clarify this important aspect of the manuscript. Yes, the distribution were checked before carrying out the analysis. We would like to clarify that as the study is carried out in the public schools, we mentioned they mostly cater to the lower socio economic strata of the population not that the most adolescents were from lower SES. We had checked the normality of socio-economic status (SES) of the participants before carrying out the formal analysis, which was normally distributed as well. 

4. There are some findings that seem inconsistent. The discussion mentions that the prevalence of overweight and obesity was 12% among girls and 9% in boys (Line 278), whereas Figure 3 has overweight at 27 and 22% respectively. Further according to Figure 2 only 13% and 6% of girls and boys respectively have adequate energy intake which is not consistent with finding of overweight and obesity. The age distribution mentioned 10-16 is quite broad for a single grade. Is this expected and accurate? Does this impact the results?

We thank the reviewer for the opportunity to clarify these points. In line 278, we provided the proportions of obesity among girls and boys, and we have revised the sentence to improve clarity. Regarding Figure 3, the proportion shown represents the combined overweight and obese categories for both girls and boys.

We acknowledge that in Figure 2, only 13% of girls and 6% of boys had adequate energy intake. However, we would like to clarify that the criteria for assessing energy intake and obesity were different. Energy intake was assessed using 24-hour dietary recalls, whereas overweight and obesity were assessed using objective measures, such as BMI. Additionally, obesity is influenced by multiple factors beyond energy intake, including a high carbohydrate diet (as shown in Figure 2), particularly refined carbs, genetics, and a lack of physical activity.

We also agree that the age distribution was broad, but the mean and median age of the participating adolescents was 13 years, which was expected. We checked the distribution of age categories before the analysis, and it was found to be normally distributed.

5. The manuscript needs a full revision and copy editing to weed out grammatical and typographical errors. I've mentioned a few examples below

Thank you. All your suggestions are valuable and have helped us improve our manuscript. As suggested we have conducted a thorough revision and copy-editing process to address grammatical and typographical errors throughout the document.

a. Please consider the statement " Adolescents were deficient in most macro and micro-nutrients, such as energy, fats, fibre, iron, zinc, iodine, riboflavin, and vitamins B-6 and B-12" within the results section in the abstract. Is energy considered a macronutrient? If not, the sentence should be rephrased.

Thank you for your valuable feedback. We have revised the statement. 

b. Some sentences have confusing syntax and need to be rephrased: Example line 29 "Owing to expensive, lifelong treatments and loss of productive life years due to the increasing premature deaths and disabilities related to various chronic diseases in India, it is recommended to focus more on promotion and prevention than curative measures."

Thank you, we have revised the line to make it more clear. 

c. The term "normal abdominal obesity" has been used multiple times in the manuscript. Normal obesity is an oxymoron. please use a more accurate term.

Thank you for bringing this to our notice. We have used the more accurate term of ‘in-range abdominal obesity.’ 

d. Line 292 on Page 14. "The mean WHR of the boys and girls were below the WHR ratio cut-offs of 0.85 and 0.9 respectively....". When using the term "respectively" the two sets of nouns should be in the same order. This sentence implies that the cutoff for boys is 0.85 and for girls 0.9.

Thank you for getting this to our notice. We have revised the sentence based on the suggestion. 

e. Figure 3 says severe acute malnutrition by MUAC and the cut off in legend is MUAC≥ 23/22 cms. The symbol "≥" connotes "greater than" and is inaccurately used here.

Thank you. We have made the correction.

---

## [Editor Report · Decision Letter 3]

11 Dec 2024

Nutritional Assessment of Adolescents: a Cross-sectional Study from Public Schools of North India

PONE-D-23-18349R3

Dear Dr. Kaur

We’re pleased to inform you that your manuscript has been judged scientifically suitable for publication and will be formally accepted for publication once it meets all outstanding technical requirements.

Kind regards,

George Kuryan

Academic Editor

PLOS ONE
---

## [Editor Report · Acceptance letter]

22 Dec 2024

PONE-D-23-18349R3 

PLOS ONE

Dear Dr. Kaur, 

I'm pleased to inform you that your manuscript has been deemed suitable for publication in PLOS ONE. Congratulations! Your manuscript is now being handed over to our production team.

Kind regards, 

on behalf of

Professor George Kuryan 

Academic Editor

PLOS ONE